# Changes in the Seroprevalence of *Helicobacter pylori* among the Lithuanian Medical Students over the Last 25 Years and Its Relation to Dyspeptic Symptoms

**DOI:** 10.3390/medicina57030254

**Published:** 2021-03-09

**Authors:** Ieva Renata Jonaityte, Eglė Ciupkeviciene, Paulius Jonaitis, Juozas Kupcinskas, Janina Petkeviciene, Laimas Jonaitis

**Affiliations:** 1Faculty of Medicine, Medical Academy, Lithuanian University of Health Sciences, A. Mickeviciaus str. 9, 44307 Kaunas, Lithuania; ieva.renata.jonaityte@stud.lsmu.lt; 2Health Research Institute, Faculty of Public Health, Lithuanian University of Health Sciences, Tilzes str. 18, 44307 Kaunas, Lithuania; egle.ciupkeviciene@lsmuni.lt (E.C.); janina.petkeviciene@lsmuni.lt (J.P.); 3Department of Gastroenterology, Lithuanian University of Health Sciences, Eiveniu str. 2, 50009 Kaunas, Lithuania; pjonaitis95@gmail.com (P.J.); juozas.kupcinskas@lsmuni.lt (J.K.)

**Keywords:** prevalence, *Helicobacter pylori*, *H. pylori* diagnostics, dyspepsia symptoms

## Abstract

*Background and Objectives*: The prevalence of *Helicobacter pylori* infection is decreasing in the Western world, while remaining high in developing countries. There is limited up-to-date information about the prevalence of *H. pylori* in Central and Eastern Europe. The aim of our study was to assess the seroprevalence of *H. pylori* and its trend over the past 25 years among students of the Lithuanian University of Health Sciences (LUHS) and to assess its relation to dyspeptic symptoms. *Materials and Methods*: In the years 1995, 2012, 2016 and 2020, students from Medical and Nursing Faculties of LUHS were tested for the presence of antibodies against *H. pylori* by performing serological tests from finger capillary blood. In addition, in the years 2012, 2016 and 2020, the students completed a gastrointestinal symptom rating scale (GSRS) questionnaire in order to assess dyspeptic symptoms. The study population consisted of 120 students in the year 1995 (mean age—21.3 ± 1.0 years), 187 students in the year 2012 (mean age—22.4 ± 0.7 years), 262 students in the year 2016 (mean age—20.4 ± 1.0 years) and 148 students in the year 2020 (mean age—20.4 ± 1.7 years). *Results*: The seroprevalence for *H. pylori* was positive in 62 (51.7%) students in 1995, in 57 (30.4%) students in 2012, in 69 (26.3%) students in 2016 and in 21 (14.2%) students in 2020. The statistically significant difference was found between all study years, except between 2012 and 2016. There were no significant differences in frequency and intensity of upper dyspeptic symptoms between *H. pylori* positive and negative students. *Conclusions*: Over the last 25 years the seroprevalence of *H. pylori* among students of LUHS has decreased significantly. No consistent differences in dyspeptic symptoms among *H. pylori* positive and negative subgroups were found.

## 1. Introduction

*Helicobacter pylori* (*H. pylori*) is the main cause of chronic gastritis and the main etiopathogenetic factor in the development of peptic ulcer disease and gastric cancer. In addition, this bacterium is associated with mucosa-associated lymphoid tissue lymphoma and hyperplastic gastric polyps [1,2,3,4]. In the year 1994, the World Health Organization classified *H. pylori* as a definite (Class I) carcinogen [5].

Even though the prevalence of *H. pylori* is decreasing worldwide, approximately 50% of the world’s population is still infected with *H. pylori* [6,7]. A wide systematic review and meta-analysis by Hooi et al., concluded that the prevalence of *H. pylori* ranges from 24.4% in the Oceania region to 70.1% in Africa. The authors stated that the prevalence of *H. pylori* in Western Europe is 34.3%; however, there is an obvious lack of data from Eastern and Central Europe [8]. New *H. pylori* epidemiology reviews are being published each year since 2003, and a trend of continuously decreasing prevalence of this bacterium has been reported from different areas around the globe [9,10].

The prevalence of *H. pylori* infection is significantly higher in Eastern and Southern European countries than in Western Europe [11]. According to various classifications, Lithuania belongs to the Eastern or Northern European region. There is a very limited number of studies investigating the prevalence of *H. pylori* in Eastern and Central European countries. There are no high-quality epidemiological *H. pylori* studies in Lithuania as well. Some investigations of smaller extent have shown the prevalence of *H. pylori* ranging from 36% in children [12] and up to 65.9% in adults in the year 2013 [13]. In addition, *H. pylori* was detected in 70% of Lithuanian patients above 55 years old with dyspeptic symptoms in 2006 [14].

The relation of *H. pylori* and dyspeptic symptoms has been discussed very extensively, and the results are controversial [15,16]. Some data indicate that there is a clear relation of symptoms and *H. pylori* gastritis: many dyspeptic symptoms are more prevalent and more intense in the *H. pylori*-positive population [17,18]. Other data do not support these findings [19].

We could hardly find the studies investigating the prevalence of *H. pylori* in similar populations (similar cohort) of the same age (~20–25 years old) in different time-points. Therefore, the aim of our research was to assess the seroprevalence of *H. pylori* and its trend over the past 25 years among the students of Lithuanian University of Health Sciences (LUHS) and to assess its relation to the dyspeptic symptoms.

## 2. Materials and Methods

### 2.1. Study Setting and Ethics

The study was performed in Lithuanian University of Health Sciences (LUHS) over the 25-year period from 1995 to 2020. The study was approved by the Bioethics Center of LUHS.

### 2.2. Study Participants

Students from the 1st to the 4th study years of the Medical and Nursing Faculties of LUHS participated in the study in 1995, 2012, 2016 and 2020. The student groups (approximately 10 students in the group) were randomly selected from the group list. All students of selected groups attending the classes on the study day were invited to participate. The demographic characteristics of the participants are presented in Table 1.

### 2.3. Methods

After signing informed consent forms, the students were tested for the presence of antibodies against *H. pylori* by performing serological tests from finger capillary blood. The serological “Helisal One-Step (Cortecs Diagnostics)” test was used in the year 1995, and “SureScreen Diagnostics Ltd.” tests were used in the years 2012, 2016 and 2020. The sensitivity and specificity of the “Helisal One-Step” test is 84% and 78% respectively [20]. According to the manufacturer, the sensitivity of the “SureScreen Diagnostics Ltd.” test is 87% and the specificity is 86% [21]. All of the tests were performed according to the given manufacturer’s instructions.

Additionally, in the years 2012, 2016 and 2020, the students completed a gastrointestinal symptom rating scale (GSRS) questionnaire in order to assess dyspeptic symptoms. The GSRS has a seven-point graded Likert-type scale where 0 represents absence of symptoms and 6 represents very troublesome symptoms. The intensity of upper gastrointestinal tract dyspeptic symptoms, such as epigastric pain or discomfort, heartburn, regurgitation, hunger-like pain, nausea, borborygmus, epigastric fullness and belching, were evaluated. All of the questionnaires were filled in anonymously.

### 2.4. Statistical Analysis

Statistical analysis was performed using IBM SPSS Statistics 25.0 and Microsoft Office Excel. Z-test with Bonferroni corrections was used for pairwise comparison of the prevalence of *H. pylori* infection between study years. χ^2^ test was used to determine the association between *H. pylori* infection and dyspeptic symptoms. The Mann–Whitney test was applied for comparing distributions of the intensity of the symptoms. The selected level of statistical significance was *p* < 0.05.

## 3. Results

### 3.1. Seroprevalence of H. pylori

The serological test for the presence of antibodies against *H. pylori* was positive in 62 (51.7%, 95% CI: 42–61%) students in 1995, in 57 (30.4%, 95% CI: 24–37%) students in 2012, in 69 (26.3%, 95% CI: 21–32%) students in 2016 and in 21 (14.2%, 95% CI: 9–20%) students in 2020 (Figure 1). The statistically significant difference (*p* < 0.05) in the prevalence of *H. pylori* was found between all the study years, except between 2012 and 2016.

The seroprevalence for *H. pylori* was found in 44 (48.4%) female and 16 (55.2%) male students in year 1995; in 40 (29.6%) female and 17 (32.7%) male students in year 2012; in 46 (66.7%) female and 23 (33.3%) male students in year 2016; in 19 (16.1%) female and 2 (7.1%) male students in year 2020. No significant difference in seroprevalence between genders was observed (*p* > 0.05).

### 3.2. H. pylori and Dyspeptic Symptoms

The main results regarding the relation between *H. pylori* status and upper gastrointestinal tract dyspeptic symptoms are presented in Table 2 and Table 3.

There were no significant differences in the prevalence and intensity of dyspeptic symptoms between *H. pylori*-positive and *H. pylori*-negative students in all study years, except for the frequency of belching in year 2020. Belching was present in 6 (28.6%) *H. pylori*-positive and in 66 (53.2%) *H. pylori*-negative students (*p* = 0.037).

## 4. Discussion

Our results indicate the obvious decline in the seroprevalence of *H. pylori* among the students of LUHS during the past 25 years. There is a clear lack of large-scale epidemiological data on the prevalence of *H. pylori* in neighboring countries as well as in whole Eastern–Central European region. Epidemiological studies in recent years have shown the different prevalence of *H. pylori* in the neighboring countries [22,23,24,25,26,27]. It has been stated that the prevalence of *H. pylori* among Latvian adults (*n* = 3564) was 79.2% in the year 2011 [22], and in fact there was no significant decrease of this infection among Latvian children during a 10-year period (2000–2010) [23]. A population study (*n* = 1461) was performed in Estonia in the year 1993 and revealed seroprevalence of *H. pylori* infection in 87% of the participants [24]. More recent studies have shown the prevalence of *H. pylori* at 69% in Tartu’s population [14] and 64.7% among the Estonian bariatric surgery patients in the year 2018 [25]. In Estonian children, the *H. pylori* seroprevalence rate was 42% in 1991 and 28.1% in 2002 [25]. Review articles [11] and other studies concluded that the prevalence of *H. pylori* ranges from 13% in children [28] to 65.6% in adults [26] in Russia, and is about 32% in Hungary [26], 23.5% in Czech Republic, 35.8% in Poland [10] and 40.8% in Romania [27].

There are very few studies available, testing participants of the similar cohorts (similar age groups (~20–25 years old) and of the same contingent) as our current research [26,29,30,31,32,33]. It has been stated that the prevalence of *H. pylori* was 23.6% among junior high school students in Poland [29], 44.1% in the age group of 18–24 years in Russia [26], ranges from 7% up to 15% in Japan [34], 14.3% in China [30], 35% among Saudi Arabia medical students [31], 68% among Yemen medical students [32], 54.7% among Taiwan high-school students and 42% in the young Israeli population [33]. Most of the above-mentioned studies have analyzed and compared data from various years and have clearly stated that the prevalence of *H. pylori* infection is decreasing. When compared to the studies on the same age groups in different countries, we can state that the prevalence of *H. pylori* in Lithuania is similar to Poland, Japan or China [29,30,34].

One of the main advantages of our research is the fact that all of the four groups of patients from different time periods were tested using the same methodology (finger capillary blood serology) and evaluated from a similar cohort (populations of similar mean age in the same university). Such studies are hardly available in other countries.

However, there are some drawbacks in our research, which need to be considered. We have to recognize that the cohorts of the four analyzed studies are not large enough to evaluate the general epidemiological situation in Lithuania, but at least we can make an assumption that it should correspond to the situation in other countries. In addition, the noninvasive diagnosis of *H. pylori* serologic tests from finger capillary blood has been used. The current edition of Maastricht V/Florence guidelines on the management of *H. pylori* infection, which is being used in European countries, recommended a urea breath test (UBT) or a monoclonal stool antigen (SAT) test for the noninvasive diagnosis of *H. pylori*, and rapid serology tests are not the best choice. However, the same guidelines state that validated serological tests with the sensitivity and specificity above 90% can be used for the noninvasive diagnostics [2].

In general, we did not find significant differences in the prevalence and intensity of dyspeptic symptoms among *H. pylori*-positive and -negative students, except for the symptom of belching, which was more prevalent in *H. pylori*-negative students in the year 2020. We would like to note that the average intensities of symptoms were quite low (1–2 points), which means that the symptoms were not really bothering and were very accidental. We assume that it is the reason why there were no differences between *H. pylori*-positive and -negative groups. In studies that reported the relation between *H. pylori* infection and dyspeptic symptoms, usually the patients with bothering dyspeptic symptoms were investigated [17,35].

## 5. Conclusions

To conclude our study, the seroprevalence for *Helicobacter pylori* was established in 14.2% of students of Lithuanian University of Health Sciences in the year 2020. Over the past 25 years, the seroprevalence of *H. pylori* among the students of LUHS has decreased significantly. No consistent differences in dyspeptic symptoms among *H. pylori*-seropositive and -negative subgroups were found.

## Figures and Tables

**Figure 1 medicina-57-00254-f001:**
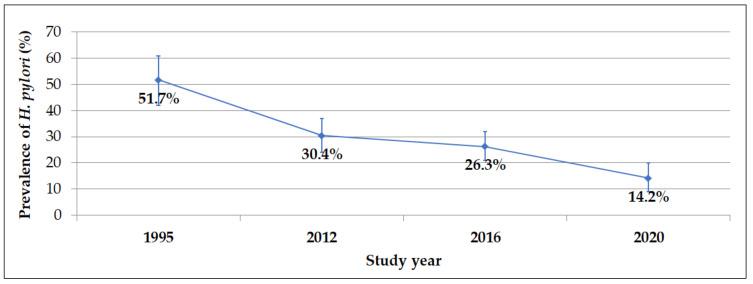
Changes in the seroprevalence of *H. pylori* over the last 25 years.

**Table 1 medicina-57-00254-t001:** Main demographic characteristics of investigated students.

	Study Year			
Characteristic	1995	2012	2016	2020
	*n* = 120	*n* = 187	*n* = 262	*n* = 148
Female, *n* (%)	91 (76%)	135 (72%)	183 (70%)	120 (81%)
Male, *n* (%)	29 (24%)	52 (28%)	78 (30%)	28 (19%)
Age, mean ± SD (years)	21.3 ± 1.0	22.4 ± 0.7	20.4 ± 1.0	20.4 ± 1.7

SD, standard deviation.

**Table 2 medicina-57-00254-t002:** Prevalence of upper gastrointestinal tract symptoms in *H. pylori*-seropositive (HP+) and *H. pylori*-seronegative (HP-) students.

				Study Year					
	2012			2016			2020		
Dyspeptic Symptoms	HP+	HP-	*p*	HP+	HP-	*p*	HP+	HP-	*p*
Epigastric discomfort	47.4%	41.5%	>0.05	47.8%	40.5%	>0.05	28.6%	40.8%	>0.05
Heartburn	29.8%	22.3%	>0.05	40.6%	33.7%	>0.05	23.8%	28.0%	>0.05
Regurgitation	22.8%	16.2%	>0.05	20.3%	27.4%	>0.05	23.8%	17.6%	>0.05
Hunger-like pain	66.7%	58.5%	>0.05	58.0%	66.8%	>0.05	66.7%	65.6%	>0.05
Nausea	28.1%	30.8%	>0.05	46.4%	40.5%	>0.05	38.1%	43.2%	>0.05
Borborygmus	75.4%	76.7%	>0.05	85.5%	79.5%	>0.05	66.7%	73.6%	>0.05
Epigastric fullness	54.4%	59.2%	>0.05	63.8%	63.2%	>0.05	38.1%	58.9%	>0.05
Belching	38.6%	44.6%	>0.05	58.0%	57.4%	>0.05	28.6%	53.2%	0.037

**Table 3 medicina-57-00254-t003:** Intensity of upper gastrointestinal tract symptoms in *H. pylori*-seropositive (HP+) and *H. pylori*-seronegative (HP-) students.

				Study Year					
	2012			2016			2020		
Dyspeptic Symptoms	HP+	HP-	*p*	HP+	HP-	*p*	HP+	HP-	*p*
Epigastric discomfort	1.1 ± 1.4	0.7 ± 1.0	>0.05	2.0 ± 1.3	2.0 ± 1.5	>0.05	2.5 ± 1.5	2.1 ± 1.4	>0.05
Heartburn	0.7 ± 1.3	0.4 ± 0.9	>0.05	2.3 ± 4.1	1.8 ± 1.3	>0.05	2.0 ± 1.4	2.3 ± 1.6	>0.05
Regurgitation	0.4 ± 0.8	0.3 ± 0.8	>0.05	1.4 ± 1.0	1.5 ± 1.1	>0.05	1.4 ± 0.9	1.7 ± 1.2	>0.05
Hunger-like pain	1.5 ± 1.4	1.3 ± 1.5	>0.05	2.2 ± 1.4	2.5 ± 1.4	>0.05	1.6 ± 0.9	2.1 ± 1.1	>0.05
Nausea	0.5 ± 1.2	0.7 ± 1.2	>0.05	2.0 ± 1.2	2.0 ± 1.5	>0.05	1.6 ± 0.7	2.2 ± 1.5	>0.05
Borborygmus	1.4 ± 1.2	1.8 ± 1.5	>0.05	3.2 ± 1.6	2.9 ± 1.4	>0.05	1.9 ± 1.2	2.5 ± 1.3	>0.05
Epigastric fullness	1.1 ± 1.3	1.3 ± 1.4	>0.05	2.6 ± 1.7	2.6 ± 1.6	>0.05	2.3 ± 1.3	2.2 ± 1.3	>0.05
Belching	0.6 ± 1.0	0.8 ± 1.2	>0.05	2.2 ± 1.5	2.1 ± 1.3	>0.05	1.5 ± 0.5	1.7 ± 1.1	>0.05

## Data Availability

Not applicable.

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
