# Peer review of "Changes in the Seroprevalence of Helicobacter pylori among the Lithuanian Medical Students over the Last 25 Years and Its Relation to Dyspeptic Symptoms"

_medicina, 2021, doi:10.3390/medicina57030254_

Round 1
Reviewer 1 Report
The original article presented by the Authors, concerning the assessment of the incidence of H. pylori in Lithuanian students at different time points, is valuable and worth publishing. Below I will present some comments that, in my opinion, should be taken into account in order to improve the quality of the manuscript.
The main concern:
My biggest objection is that the Authors of the manuscript use the expression "prevalence" instead of SEROPREVALENCE. Although the difference may seem marginal, it still significantly influences the perception of the article.
As the Authors themselves admit in lines 167-174, serology is not the best method of assessing the presence of H. pylori (especially because it makes it impossible to assess whether the infection is active or is only a trace of immunological memory). One could try to indicate the prevalence of H. pylori if the serology were extended to another diagnostic method, such as antigen tests, UBT or bacterial culture – but this is not the case.
At this point I will also refer to the sentence introduced by the Authors "However, the same guidelines state that validated serological tests with the sensitivity and specificity above 90% can be used for the non-invasive diagnostics". It is true, but it is worth noting that this criterion is not met by the Authors, because the “Helisal One-Step” test has sensitivity and specificity of 84% and 78%, respectively, while the “SureScreen Diagnostics Ltd” test has sensitivity and specificity of 87% and 86%, respectively.
Proposed solution:
Despite my above argument, I still believe that the article should be published, because it carries important cognitive values. One should carefully review the entire article and replace all terms about prevalence (the one studied by the Authors of course) with “seroprevalence”.
E.g. in the title of the manuscript, the title of Figure 1, and lines: 18, 31, 124, 133, and 186, and additionally please change the following sentences:
- “H. pylori test was positive in 62 (51.7%) students in 1995, in 57 (30.4%) students in 2012, in 69 (26.3%) students in 2016 and in 21 (14.2%) students in 2020.” -> The seroprevalence for H. pylori was positive in 62 (51.7%) students in 1995, in 57 (30.4%) students in 2012, in 69 (26.3%) students in 2016 and in 21 (14.2%) students in 2020. [lines 27-28]
- “H. pylori was found in 44 (48.4%) female and 16 (55.2%) male students in year 1995 …” -> The seroprevalence for H. pylori was found in 44 (48.4%) female and 16 (55.2%) male students in year 1995 … [line 112]
- “To conclude our study, Helicobacter pylori was established in 14.2% of students of Lithuanian University of Health Sciences in the year 2020. Over the last 25 years, the prevalence of H. pylori among the students of LUHS has decreased significantly. No consistent differences in dyspeptic symptoms among H. pylori positive and negative subgroups were found.” -> To conclude our study, the seroprevalence for Helicobacter pylori was established in 14.2% of students of Lithuanian University of Health Sciences in the year 2020. Over the last 25 years, the seroprevalence agaisnt H. pylori among the students of LUHS has decreased significantly. No consistent differences in dyspeptic symptoms among H. pylori seropositive and -negative subgroups were found. [Conclusions]
Additionally some minor changes:
- “over the last 25 years …” -> Over the last 25 years … [line 31]
- “and in 19 (16.1%) female and 2 (7.1%) male students in year 2020.” -> There is a mistake here because the sum of these values should be 100%, as previously indicated [line 114-115]
- modifications of the Tables 2 and 3, including: extension of tables to the entire length of the page, not only part of it (it is in accordance with the guidelines of the journal), in the top of the Tables there should be the years 2012, 2016 and 2020 INSTEAD OF 1995, 2012 and 2016; and additionally please write > 0.05 (in the p-value column) only one for an entire column -> more easy to read
- “One of the main advantages of our research is the fact that all of the four studies of different years used the same methods (serological tests) and evaluated the similar cohort (populations of similar mean age in the same university). Such studies are hardly available in other countries. Another important advantage is that the same method of testing (finger capillary blood serology) has been used in all study cohorts in different years.” -> One of the main advantages of our research is the fact that all of the four groups of patients from different time periods were tested using the same methodology (finger capillary blood serology) and evaluated the similar cohort (populations of similar mean age in the same university). Such studies are hardly available in other countries. [lines 159-163]
Author Response
Response to Reviewer 1 comments
We would like to thank you for your effort and time to read and revise our article. All your suggestions were excellent and helped us to improve our manuscript. We hope that we have successfully addressed all of the concerns raised. Our detailed responses to the comments and the description of changes we have made to the manuscript are provided below.
Point 1: My biggest objection is that the Authors of the manuscript use the expression "prevalence" instead of SEROPREVALENCE. Although the difference may seem marginal, it still significantly influences the perception of the article.
Response 1: Following this excellent advice of the reviewer, the changes have been made in the manuscript. The “prevalence” has been changed to “seroprevalence” in the title of the manuscript, the title of Figure 1, lines: 18, 31, 133, and 186, and additionally in all chapters and sentences where it is applicable.
For example,
- The title has been changed to “Changes in the Seroprevalence of Helicobacter pylori Among........“
- “Therefore, the aim of our research was to assess the seroprevalence of pylori.....“
- Title of Fig.1 is: „Figure 1. Changes in the seroprevalence of H. pylori over the last 25 years”
- Line 18: “…was to assess the seroprevalence of H. pylori and its trend over the last 25 years among students of...“
- Line 31: “Over the last 25 years the seroprevalence of H. pylori among students of LUHS has decreased....“
- Line 133: “Our results indicate the obvious decline in the seroprevalence of H. pylori among the...“
- Line 186-187: “Over the last 25 years, the seroprevalence of H. pylori among the students of LUHS has decreased significantly.”
- And everywhere throughout the manuscript.
The corrections that have been made according to the reviewer’s suggestions:
Suggestion 1. “H. pylori test was positive in 62 (51.7%) students in 1995, in 57 (30.4%) students in 2012, in 69 (26.3%) students in 2016 and in 21 (14.2%) students in 2020.” -> The seroprevalence for H. pylori was positive in 62 (51.7%) students in 1995, in 57 (30.4%) students in 2012, in 69 (26.3%) students in 2016 and in 21 (14.2%) students in 2020. [lines 27-28]
Response: Now it is corrected as the reviewer suggested.
Suggestion 2. - “H. pylori was found in 44 (48.4%) female and 16 (55.2%) male students in year 1995 …” -> The seroprevalence for H. pylori was found in 44 (48.4%) female and 16 (55.2%) male students in year 1995 … [line 112]
Response: Now it is corrected as the reviewer suggested.
Suggestion 3. “To conclude our study, Helicobacter pylori was established in 14.2% of students of Lithuanian University of Health Sciences in the year 2020. Over the last 25 years, the prevalence of H. pylori among the students of LUHS has decreased significantly. No consistent differences in dyspeptic symptoms among H. pylori positive and negative subgroups were found.” -> To conclude our study, the seroprevalence for Helicobacter pylori was established in 14.2% of students of Lithuanian University of Health Sciences in the year 2020. Over the last 25 years, the seroprevalence of H. pylori among the students of LUHS has decreased significantly. No consistent differences in dyspeptic symptoms among H. pylori seropositive and -negative subgroups were found.
Response: Now it is corrected as the reviewer suggested.
Point 2: line 31 -“over the last 25 years …”-> Over the last 25 years
Response 2: Corrected.
Now the sentence is: “Conclusions: Over the last 25 years the seroprevalence of H. pylori among students of LUHS has decreased significantly.“
Point 3: “and in 19 (16.1%) female and 2 (7.1%) male students in year 2020.” -> There is a mistake here because the sum of these values should be 100%, as previously indicated [line 114-115].
Response 3:
We would like to kindly notify, that there is no necessity to be 100% as it reflects the seroprevalence among females and males in the given year.
For the better understanding of this data we slightly changed the paragraph. Now it is:
The seroprevalence for H. pylori was found in 44 (48.4%) female and 16 (55.2%) male students in year 1995; in 40 (29.6%) female and 17 (32.7%) male students in year 2012; in 46 (66.7%) female and 23 (33.3%) male students in year 2016; in 19 (16.1%) female and 2 (7.1%) male students in year 2020. No significant difference in seroprevalence between genders was observed (p>0.05).
Point 4: Modifications of the Tables 2 and 3, including: extension of tables to the entire length of the page, not only part of it (it is in accordance with the guidelines of the journal), in the top of the Tables there should be the years 2012, 2016 and 2020 INSTEAD OF 1995, 2012 and 2016; and additionally please write > 0.05 (in the p-value column) only one for an entire column -> more easy to read
Response 4:
It is corrected according to the reviewer’s suggestions: the tables were extended to the entire length of the page and the years were corrected.
We would like to notify that we could not find a technical solution how to leave only one p-value for the entire column because in the Table 2 the p-value for “belching” was different.
The updated tables are below:
Table 2. Prevalence of upper gastrointestinal tract symptoms in H. pylori-seropositive and H. pylori-seronegative students
|
|
Study year |
|
|||||||
|
|
2012 |
2016 |
2020 |
||||||
|
Dyspeptic symptoms |
HP+ |
HP- |
p |
HP+ |
HP- |
p |
HP+ |
HP- |
p |
|
Epigastric discomfort |
47.4% |
41.5% |
>0.05 |
47.8% |
40.5% |
>0.05 |
28.6% |
40.8% |
>0.05 |
|
Heartburn |
29.8% |
22.3% |
>0.05 |
40.6% |
33.7% |
>0.05 |
23.8% |
28.0% |
>0.05 |
|
Regurgitation |
22.8% |
16.2% |
>0.05 |
20.3% |
27.4% |
>0.05 |
23.8% |
17.6% |
>0.05 |
|
Hunger-like pain |
66.7% |
58.5% |
>0.05 |
58.0% |
66.8% |
>0.05 |
66.7% |
65.6% |
>0.05 |
|
Nausea |
28.1% |
30.8% |
>0.05 |
46.4% |
40.5% |
>0.05 |
38.1% |
43.2% |
>0.05 |
|
Borborygmus |
75.4% |
76.7% |
>0.05 |
85.5% |
79.5% |
>0.05 |
66.7% |
73.6% |
>0.05 |
|
Epigastric fullness |
54.4% |
59.2% |
>0.05 |
63.8% |
63.2% |
>0.05 |
38.1% |
58.9% |
>0.05 |
|
Belching |
38.6% |
44.6% |
>0.05 |
58.0% |
57.4% |
>0.05 |
28.6% |
53.2% |
0.037 |
Table 3. Intensity of upper gastrointestinal tract symptoms in H. pylori-seropositive and H. pylori-seronegative
students
|
|
Study year |
|
|||||||
|
|
2012 |
2016 |
2020 |
||||||
|
Dyspeptic symptoms |
HP+ |
HP- |
p |
HP+ |
HP- |
p |
HP+ |
HP- |
p |
|
Epigastric discomfort |
1.1±1.4 |
0.7±1.0 |
>0.05 |
2.0±1.3 |
2.0±1.5 |
>0.05 |
2.5±1.5 |
2.1±1.4 |
>0.05 |
|
Heartburn |
0.7±1.3 |
0.4±0.9 |
>0.05 |
2.3±4.1 |
1.8±1.3 |
>0.05 |
2.0±1.4 |
2.3±1.6 |
>0.05 |
|
Regurgitation |
0.4±0.8 |
0.3±0.8 |
>0.05 |
1.4±1.0 |
1.5±1.1 |
>0.05 |
1.4±0.9 |
1.7±1.2 |
>0.05 |
|
Hunger-like pain |
1.5±1.4 |
1.3±1.5 |
>0.05 |
2.2±1.4 |
2.5±1.4 |
>0.05 |
1.6±0.9 |
2.1±1.1 |
>0.05 |
|
Nausea |
0.5±1.2 |
0.7±1.2 |
>0.05 |
2.0±1.2 |
2.0±1.5 |
>0.05 |
1.6±0.7 |
2.2±1.5 |
>0.05 |
|
Borborygmus |
1.4±1.2 |
1.8±1.5 |
>0.05 |
3.2±1.6 |
2.9±1.4 |
>0.05 |
1.9±1.2 |
2.5±1.3 |
>0.05 |
|
Epigastric fullness |
1.1±1.3 |
1.3±1.4 |
>0.05 |
2.6±1.7 |
2.6±1.6 |
>0.05 |
2.3±1.3 |
2.2±1.3 |
>0.05 |
|
Belching |
0.6±1.0 |
0.8±1.2 |
>0.05 |
2.2±1.5 |
2.1±1.3 |
>0.05 |
1.5±0.5 |
1.7±1.1 |
>0.05 |
Point 5: “One of the main advantages of our research is the fact that all of the four studies of different years used the same methods (serological tests) and evaluated the similar cohort (populations of similar mean age in the same university). Such studies are hardly available in other countries. Another important advantage is that the same method of testing (finger capillary blood serology) has been used in all study cohorts in different years.” -> One of the main advantages of our research is the fact that all of the four groups of patients from different time periods were tested using the same methodology (finger capillary blood serology) and evaluated the similar cohort (populations of similar mean age in the same university). Such studies are hardly available in other countries. [lines 159-163]
Response 5:
The paragraph has been corrected according to the reviewer’s suggestion:
„One of the main advantages of our research is the fact that all of the four groups of patients from different time periods were tested using the same methodology (finger capillary blood serology) and evaluated the similar cohort (populations of similar mean age in the same university). Such studies are hardly available in other countries.“
Point 6: As the Authors themselves admit in lines 167-174, serology is not the best method of assessing the presence of H. pylori (especially because it makes it impossible to assess whether the infection is active or is only a trace of immunological memory). One could try to indicate the prevalence of H. pylori if the serology were extended to another diagnostic method, such as antigen tests, UBT or bacterial culture – but this is not the case.
Response 6:
We totally agree with the reviewer regarding this issue, but at the moment we started our investigation of the students the serology was still widely accepted and recommended method. We addressed this issue as a drawback of our study in the discussion.
Point 7: At this point I will also refer to the sentence introduced by the Authors "However, the same guidelines state that validated serological tests with the sensitivity and specificity above 90% can be used for the non-invasive diagnostics". It is true, but it is worth noting that this criterion is not met by the Authors, because the “Helisal One-Step” test has sensitivity and specificity of 84% and 78%, respectively, while the “SureScreen Diagnostics Ltd” test has sensitivity and specificity of 87% and 86%, respectively.
Response 7:
Again, this is a totally correct notification. The recommendations to use validated and highly accurate tests were accepted in the recent Maastricht/Florence consensuses. And it is usually the laboratory based tests, not the bed-side quick tests (as we used in our research). As we performed our first series, the quick serological tests were still acceptable for the primary diagnostics. The sensitivity and specificity of tests used in our research is not far from recommended 90%.
Moreover, the quick bed-side serological tests are still used in many countries. If the patient has not been treated against Helicobacter pylori infection – the positive test is probably accurate enough to confirm the H. pylori infection.
Response to Reviewer 1 comments
We would like to thank you for your effort and time to read and revise our article. All your suggestions were excellent and helped us to improve our manuscript. We hope that we have successfully addressed all of the concerns raised. Our detailed responses to the comments and the description of changes we have made to the manuscript are provided below.
Point 1: My biggest objection is that the Authors of the manuscript use the expression "prevalence" instead of SEROPREVALENCE. Although the difference may seem marginal, it still significantly influences the perception of the article.
Response 1: Following this excellent advice of the reviewer, the changes have been made in the manuscript. The “prevalence” has been changed to “seroprevalence” in the title of the manuscript, the title of Figure 1, lines: 18, 31, 133, and 186, and additionally in all chapters and sentences where it is applicable.
For example,
- The title has been changed to “Changes in the Seroprevalence of Helicobacter pylori Among........“
- “Therefore, the aim of our research was to assess the seroprevalence of pylori.....“
- Title of Fig.1 is: „Figure 1. Changes in the seroprevalence of H. pylori over the last 25 years”
- Line 18: “…was to assess the seroprevalence of H. pylori and its trend over the last 25 years among students of...“
- Line 31: “Over the last 25 years the seroprevalence of H. pylori among students of LUHS has decreased....“
- Line 133: “Our results indicate the obvious decline in the seroprevalence of H. pylori among the...“
- Line 186-187: “Over the last 25 years, the seroprevalence of H. pylori among the students of LUHS has decreased significantly.”
- And everywhere throughout the manuscript.
The corrections that have been made according to the reviewer’s suggestions:
Suggestion 1. “H. pylori test was positive in 62 (51.7%) students in 1995, in 57 (30.4%) students in 2012, in 69 (26.3%) students in 2016 and in 21 (14.2%) students in 2020.” -> The seroprevalence for H. pylori was positive in 62 (51.7%) students in 1995, in 57 (30.4%) students in 2012, in 69 (26.3%) students in 2016 and in 21 (14.2%) students in 2020. [lines 27-28]
Response: Now it is corrected as the reviewer suggested.
Suggestion 2. - “H. pylori was found in 44 (48.4%) female and 16 (55.2%) male students in year 1995 …” -> The seroprevalence for H. pylori was found in 44 (48.4%) female and 16 (55.2%) male students in year 1995 … [line 112]
Response: Now it is corrected as the reviewer suggested.
Suggestion 3. “To conclude our study, Helicobacter pylori was established in 14.2% of students of Lithuanian University of Health Sciences in the year 2020. Over the last 25 years, the prevalence of H. pylori among the students of LUHS has decreased significantly. No consistent differences in dyspeptic symptoms among H. pylori positive and negative subgroups were found.” -> To conclude our study, the seroprevalence for Helicobacter pylori was established in 14.2% of students of Lithuanian University of Health Sciences in the year 2020. Over the last 25 years, the seroprevalence of H. pylori among the students of LUHS has decreased significantly. No consistent differences in dyspeptic symptoms among H. pylori seropositive and -negative subgroups were found.
Response: Now it is corrected as the reviewer suggested.
Point 2: line 31 -“over the last 25 years …”-> Over the last 25 years
Response 2: Corrected.
Now the sentence is: “Conclusions: Over the last 25 years the seroprevalence of H. pylori among students of LUHS has decreased significantly.“
Point 3: “and in 19 (16.1%) female and 2 (7.1%) male students in year 2020.” -> There is a mistake here because the sum of these values should be 100%, as previously indicated [line 114-115].
Response 3:
We would like to kindly notify, that there is no necessity to be 100% as it reflects the seroprevalence among females and males in the given year.
For the better understanding of this data we slightly changed the paragraph. Now it is:
The seroprevalence for H. pylori was found in 44 (48.4%) female and 16 (55.2%) male students in year 1995; in 40 (29.6%) female and 17 (32.7%) male students in year 2012; in 46 (66.7%) female and 23 (33.3%) male students in year 2016; in 19 (16.1%) female and 2 (7.1%) male students in year 2020. No significant difference in seroprevalence between genders was observed (p>0.05).
Point 4: Modifications of the Tables 2 and 3, including: extension of tables to the entire length of the page, not only part of it (it is in accordance with the guidelines of the journal), in the top of the Tables there should be the years 2012, 2016 and 2020 INSTEAD OF 1995, 2012 and 2016; and additionally please write > 0.05 (in the p-value column) only one for an entire column -> more easy to read
Response 4:
It is corrected according to the reviewer’s suggestions: the tables were extended to the entire length of the page and the years were corrected.
We would like to notify that we could not find a technical solution how to leave only one p-value for the entire column because in the Table 2 the p-value for “belching” was different.
The updated tables are below:
Table 2. Prevalence of upper gastrointestinal tract symptoms in H. pylori-seropositive and H. pylori-seronegative students
|
|
Study year |
|
|||||||
|
|
2012 |
2016 |
2020 |
||||||
|
Dyspeptic symptoms |
HP+ |
HP- |
p |
HP+ |
HP- |
p |
HP+ |
HP- |
p |
|
Epigastric discomfort |
47.4% |
41.5% |
>0.05 |
47.8% |
40.5% |
>0.05 |
28.6% |
40.8% |
>0.05 |
|
Heartburn |
29.8% |
22.3% |
>0.05 |
40.6% |
33.7% |
>0.05 |
23.8% |
28.0% |
>0.05 |
|
Regurgitation |
22.8% |
16.2% |
>0.05 |
20.3% |
27.4% |
>0.05 |
23.8% |
17.6% |
>0.05 |
|
Hunger-like pain |
66.7% |
58.5% |
>0.05 |
58.0% |
66.8% |
>0.05 |
66.7% |
65.6% |
>0.05 |
|
Nausea |
28.1% |
30.8% |
>0.05 |
46.4% |
40.5% |
>0.05 |
38.1% |
43.2% |
>0.05 |
|
Borborygmus |
75.4% |
76.7% |
>0.05 |
85.5% |
79.5% |
>0.05 |
66.7% |
73.6% |
>0.05 |
|
Epigastric fullness |
54.4% |
59.2% |
>0.05 |
63.8% |
63.2% |
>0.05 |
38.1% |
58.9% |
>0.05 |
|
Belching |
38.6% |
44.6% |
>0.05 |
58.0% |
57.4% |
>0.05 |
28.6% |
53.2% |
0.037 |
Table 3. Intensity of upper gastrointestinal tract symptoms in H. pylori-seropositive and H. pylori-seronegative
students
|
|
Study year |
|
|||||||
|
|
2012 |
2016 |
2020 |
||||||
|
Dyspeptic symptoms |
HP+ |
HP- |
p |
HP+ |
HP- |
p |
HP+ |
HP- |
p |
|
Epigastric discomfort |
1.1±1.4 |
0.7±1.0 |
>0.05 |
2.0±1.3 |
2.0±1.5 |
>0.05 |
2.5±1.5 |
2.1±1.4 |
>0.05 |
|
Heartburn |
0.7±1.3 |
0.4±0.9 |
>0.05 |
2.3±4.1 |
1.8±1.3 |
>0.05 |
2.0±1.4 |
2.3±1.6 |
>0.05 |
|
Regurgitation |
0.4±0.8 |
0.3±0.8 |
>0.05 |
1.4±1.0 |
1.5±1.1 |
>0.05 |
1.4±0.9 |
1.7±1.2 |
>0.05 |
|
Hunger-like pain |
1.5±1.4 |
1.3±1.5 |
>0.05 |
2.2±1.4 |
2.5±1.4 |
>0.05 |
1.6±0.9 |
2.1±1.1 |
>0.05 |
|
Nausea |
0.5±1.2 |
0.7±1.2 |
>0.05 |
2.0±1.2 |
2.0±1.5 |
>0.05 |
1.6±0.7 |
2.2±1.5 |
>0.05 |
|
Borborygmus |
1.4±1.2 |
1.8±1.5 |
>0.05 |
3.2±1.6 |
2.9±1.4 |
>0.05 |
1.9±1.2 |
2.5±1.3 |
>0.05 |
|
Epigastric fullness |
1.1±1.3 |
1.3±1.4 |
>0.05 |
2.6±1.7 |
2.6±1.6 |
>0.05 |
2.3±1.3 |
2.2±1.3 |
>0.05 |
|
Belching |
0.6±1.0 |
0.8±1.2 |
>0.05 |
2.2±1.5 |
2.1±1.3 |
>0.05 |
1.5±0.5 |
1.7±1.1 |
>0.05 |
Point 5: “One of the main advantages of our research is the fact that all of the four studies of different years used the same methods (serological tests) and evaluated the similar cohort (populations of similar mean age in the same university). Such studies are hardly available in other countries. Another important advantage is that the same method of testing (finger capillary blood serology) has been used in all study cohorts in different years.” -> One of the main advantages of our research is the fact that all of the four groups of patients from different time periods were tested using the same methodology (finger capillary blood serology) and evaluated the similar cohort (populations of similar mean age in the same university). Such studies are hardly available in other countries. [lines 159-163]
Response 5:
The paragraph has been corrected according to the reviewer’s suggestion:
„One of the main advantages of our research is the fact that all of the four groups of patients from different time periods were tested using the same methodology (finger capillary blood serology) and evaluated the similar cohort (populations of similar mean age in the same university). Such studies are hardly available in other countries.“
Point 6: As the Authors themselves admit in lines 167-174, serology is not the best method of assessing the presence of H. pylori (especially because it makes it impossible to assess whether the infection is active or is only a trace of immunological memory). One could try to indicate the prevalence of H. pylori if the serology were extended to another diagnostic method, such as antigen tests, UBT or bacterial culture – but this is not the case.
Response 6:
We totally agree with the reviewer regarding this issue, but at the moment we started our investigation of the students the serology was still widely accepted and recommended method. We addressed this issue as a drawback of our study in the discussion.
Point 7: At this point I will also refer to the sentence introduced by the Authors "However, the same guidelines state that validated serological tests with the sensitivity and specificity above 90% can be used for the non-invasive diagnostics". It is true, but it is worth noting that this criterion is not met by the Authors, because the “Helisal One-Step” test has sensitivity and specificity of 84% and 78%, respectively, while the “SureScreen Diagnostics Ltd” test has sensitivity and specificity of 87% and 86%, respectively.
Response 7:
Again, this is a totally correct notification. The recommendations to use validated and highly accurate tests were accepted in the recent Maastricht/Florence consensuses. And it is usually the laboratory based tests, not the bed-side quick tests (as we used in our research). As we performed our first series, the quick serological tests were still acceptable for the primary diagnostics. The sensitivity and specificity of tests used in our research is not far from recommended 90%.
Moreover, the quick bed-side serological tests are still used in many countries. If the patient has not been treated against Helicobacter pylori infection – the positive test is probably accurate enough to confirm the H. pylori infection.
Reviewer 2 Report
This manuscript by Ieva Renata Jonaityte et al. describe a simple but very interesting data focusing on the prevalence of Hp in lithuanian patients.
Methods : please indicate correctly the reference of the biological assays.
Results : Figure 1 : indicate in the same figure the evolution of the main symptoms.
Discussion : Numerous sentence lack reference. Please add.
Author Response
Response to Reviewer 2 comments
Thank you very much for reviewing our manuscript. We appreciate your comments and suggestions. We hope that we have successfully addressed all of the concerns raised and we believe that the manuscript has been substantially improved. Our detailed responses to the comments and the description of the changes we have made to the manuscript are provided below.
Point 1: Methods - please indicate correctly the reference of the biological assays.
Response 1:
We added the reference for the “SureScreen Diagnostics Ltd” test:
“According to the manufacturer, the sensitivity of the “SureScreen Diagnostics Ltd” test is 87% and the specificity is 86% [21].”
- H. pylori Rapid Test Device (Whole Blood/Serum/Plasma) Package Insert https://www.gimaitaly.com/DocumentiGIMA/Manuali/EN/M24528EN.pdf
Point 2: Results - Figure 1: indicate in the same figure the evolution of the main symptoms.
Response 2:
Dear Reviewer,
We were thinking about the comprehensive representation of the symptoms before submitting the manuscript. However, we decided not to overload the manuscript with additional figures that are not statistically significant. This was not the primary aim of our study. In case of the figure with trendlines – it is too many trendlines in one figure and it would be difficult to add numbers and SD. Overall, it would be less informative and more difficult to read the figure. We would like to stress that the average intensity of symptoms was low and was not significantly different throughout the years. Moreover, we are emphasizing the difference of symptoms between HP-seropositive and HP-seronegative groups. We did not have an intension to compare the evolution of the symptoms in different years. Therefore, we decided to represent the prevalence and intensity of symptoms in the Tables 2 and 3, as it looked most informative.
We hope you could accept our explanation and decision.
Table 2. Prevalence of upper gastrointestinal tract symptoms in H. pylori-seropositive and H. pylori-seronegative students
|
|
Study year |
|
|||||||
|
|
2012 |
2016 |
2020 |
||||||
|
Dyspeptic symptoms |
HP+ |
HP- |
p |
HP+ |
HP- |
p |
HP+ |
HP- |
p |
|
Epigastric discomfort |
47.4% |
41.5% |
>0.05 |
47.8% |
40.5% |
>0.05 |
28.6% |
40.8% |
>0.05 |
|
Heartburn |
29.8% |
22.3% |
>0.05 |
40.6% |
33.7% |
>0.05 |
23.8% |
28.0% |
>0.05 |
|
Regurgitation |
22.8% |
16.2% |
>0.05 |
20.3% |
27.4% |
>0.05 |
23.8% |
17.6% |
>0.05 |
|
Hunger-like pain |
66.7% |
58.5% |
>0.05 |
58.0% |
66.8% |
>0.05 |
66.7% |
65.6% |
>0.05 |
|
Nausea |
28.1% |
30.8% |
>0.05 |
46.4% |
40.5% |
>0.05 |
38.1% |
43.2% |
>0.05 |
|
Borborygmus |
75.4% |
76.7% |
>0.05 |
85.5% |
79.5% |
>0.05 |
66.7% |
73.6% |
>0.05 |
|
Epigastric fullness |
54.4% |
59.2% |
>0.05 |
63.8% |
63.2% |
>0.05 |
38.1% |
58.9% |
>0.05 |
|
Belching |
38.6% |
44.6% |
>0.05 |
58.0% |
57.4% |
>0.05 |
28.6% |
53.2% |
0.037 |
Table 3. Intensity of upper gastrointestinal tract symptoms in H. pylori-seropositive and H. pylori-seronegative
students
|
|
Study year |
|
|||||||
|
|
2012 |
2016 |
2020 |
||||||
|
Dyspeptic symptoms |
HP+ |
HP- |
p |
HP+ |
HP- |
p |
HP+ |
HP- |
p |
|
Epigastric discomfort |
1.1±1.4 |
0.7±1.0 |
>0.05 |
2.0±1.3 |
2.0±1.5 |
>0.05 |
2.5±1.5 |
2.1±1.4 |
>0.05 |
|
Heartburn |
0.7±1.3 |
0.4±0.9 |
>0.05 |
2.3±4.1 |
1.8±1.3 |
>0.05 |
2.0±1.4 |
2.3±1.6 |
>0.05 |
|
Regurgitation |
0.4±0.8 |
0.3±0.8 |
>0.05 |
1.4±1.0 |
1.5±1.1 |
>0.05 |
1.4±0.9 |
1.7±1.2 |
>0.05 |
|
Hunger-like pain |
1.5±1.4 |
1.3±1.5 |
>0.05 |
2.2±1.4 |
2.5±1.4 |
>0.05 |
1.6±0.9 |
2.1±1.1 |
>0.05 |
|
Nausea |
0.5±1.2 |
0.7±1.2 |
>0.05 |
2.0±1.2 |
2.0±1.5 |
>0.05 |
1.6±0.7 |
2.2±1.5 |
>0.05 |
|
Borborygmus |
1.4±1.2 |
1.8±1.5 |
>0.05 |
3.2±1.6 |
2.9±1.4 |
>0.05 |
1.9±1.2 |
2.5±1.3 |
>0.05 |
|
Epigastric fullness |
1.1±1.3 |
1.3±1.4 |
>0.05 |
2.6±1.7 |
2.6±1.6 |
>0.05 |
2.3±1.3 |
2.2±1.3 |
>0.05 |
|
Belching |
0.6±1.0 |
0.8±1.2 |
>0.05 |
2.2±1.5 |
2.1±1.3 |
>0.05 |
1.5±0.5 |
1.7±1.1 |
>0.05 |
Point 3: Discussion - Numerous sentence lack reference. Please add.
Response 3:
Dear Reviewer, of course, we may agree that there is not enough references in the “Discussion” chapter. We tried to support our main discussion points with the appropriate references. However, there is a recommendation to limit the list of references to the reasonable amount. Therefore, we could not be very extensive.
According to Your suggestion, we have added some references and added the numbers of the references to some sentences.
Probably, we may kindly add few more references, in case the reviewer could indicate the points where the references should be added? We are not sure in which parts of the “Discussion” the references are insufficient as the Reviewer did not indicate specifically.
Please find below the updated chapter “Discussion” and the reference list, with the highlighted changes
- Discussion
Our results indicate the obvious decline in the seroprevalence of H. pylori among the students of LUHS during the last 25 years. There is a clear lack of large-scale epidemiological data on the prevalence of H. pylori in neighboring countries as well as in whole Eastern-Central European region. Epidemiological studies in recent years have shown the different prevalence of H. pylori in the neighboring countries [22-25,27,28]. It has been stated that the prevalence of H. pylori among Latvian adults (n=3564) was 79.2% in the year 2011 [22] and in fact there was no significant decrease of this infection among Latvian children during a 10-year period (2000-2010) [23]. A population study (n=1461) was performed in Estonia in the year 1993 and revealed seroprevalence of H. pylori infection in 87% of the participants [24]. More recent studies have shown the prevalence of H. pylori 69% in Tartu’s population [14] and 64.7% among the Estonian bariatric surgery patients in the year 2018 [25]. In Estonian children, the H. pylori seroprevalence rate was 42% in 1991 and 28.1% in 2002 [25]. Review article [11] and other studies concluded that the prevalence of H. pylori ranges from 13% in children [26] to 65.6% in adults [27] in Russia, is around 32% in Hungary [27], 23.5% in Czech Republic, 35.8% in Poland [10] and 40.8% in Romania [28].
There are very few studies available, testing participants of the similar cohorts (similar age groups (~20-25 years old) and of the same contingent) as our current research [27,29,31-34]. It has been stated that the prevalence of H. pylori was 23.6% among junior high school students in Poland [29], 44.1% in the age group of 18-24 years in Russia [27], ranges from 7% up to 15% in Japan [30], was 14.3% in China [31], 35% among Saudi Arabia medical students [32], 68% among Yemen medical students [33], 54.7% among Taiwan high-school students and 42% in young Jewish population [34]. Most of the above-mentioned studies have analyzed and compared data from various years and have clearly stated that the prevalence of H. pylori infection is decreasing. When compared to the studies on the same age groups in different countries, we can state that the prevalence of H. pylori in Lithuania is similar to Poland, Japan or China [29-31].
One of the main advantages of our research is the fact that all of the four groups of patients from different time periods were tested using the same methodology (finger capillary blood serology) and evaluated the similar cohort (populations of similar mean age in the same university). Such studies are hardly available in other countries.
However, there are some drawbacks in our research, which need to be considered. We have to recognize, that the cohorts of the four analyzed studies are not large enough in order to evaluate the general epidemiological situation in Lithuania but at least we can make an assumption that it should correspond to the situation in other countries. In addition, the non-invasive diagnosis of H. pylori serologic tests from finger capillary blood has been used. The current edition of Maastricht V/Florence guidelines on the management of H. pylori infection, which is being used in European countries, recommended urea breath test (UBT) or monoclonal stool antigen (SAT) test for the non-invasive diagnosis of H. pylori and rapid serology tests are not the best choice. However, the same guidelines state that validated serological tests with the sensitivity and specificity above 90% can be used for the non-invasive diagnostics [2].
In general, we did not find significant differences in the prevalence and intensity of dyspeptic symptoms among H. pylori-positive and negative students, except for the symptom of belching which was more prevalent in H. pylori-negative students in the year 2020. We would like to note that the average intensities of symptoms were quite low (1-2 points), which means that the symptoms were not really bothering and were very accidental. We assume that it is the reason, why there were no differences between H. pylori-positive and negative groups. In studies, that reported the relation between H. pylori infection and dyspeptic symptoms, usually the patients with bothering dyspeptic symptoms were investigated [35,36].
- Conclusions
To conclude our study, the seroprevalence for Helicobacter pylori was established in 14.2% of students of Lithuanian University of Health Sciences in the year 2020. Over the last 25 years, the seroprevalence of H. pylori among the students of LUHS has decreased significantly. No consistent differences in dyspeptic symptoms among H. pylori seropositive and -negative subgroups were found.
References
- Wroblewski, L.E.; Peek, R.M.J.; Wilson, K.T. Helicobacter pylori and gastric cancer: factors that modulate disease risk. Clin Microbiol Rev 2010, 23, 713–739.
- Malfertheiner, P.; Megraud, F.; O’Morain, C.A.; Gisbert, J.P.; Kuipers, E.J.; Axon, A.T.; et al. Management of Helicobacter pylori infection - the Maastricht V/Florence Consensus Report. Gut 2017, 66, 6–30.
- Chey, W.D.; Leontiadis, G.I.; Howden, C.W.; Moss, S.F. Correction: ACG Clinical Guideline: Treatment of Helicobacter pylori Infection. The American journal of gastroenterology 2018, 113, 1102.
- Kato, M.; Ota, H.; Okuda, M.; Kikuchi, S.; Satoh, K.; Shimoyama, T.; et al. Guidelines for the management of Helicobacter pylori infection in Japan: 2016 Revised Edition. Helicobacter 2019, 24, e12597.
- Moss, S.F. The Clinical Evidence Linking Helicobacter pylori to Gastric Cancer. Cell Mol Gastroenterol Hepatol 2017, 3, 183–191.
- Šterbenc, A.; Jarc, E.; Poljak, M.; Homan, M. Helicobacter pylori virulence genes. World J Gastroenterol 2019, 25, 4870–4884.
- Kupcinskas, L.; Rasmussen, L.; Jonaitis, L.; Kiudelis, G.; Jørgensen, M.; Urbonaviciene, N.; et al. Evolution of Helicobacter pylori susceptibility to antibiotics during a 10-year period in Lithuania. APMIS 2013, 121, 431–436.
- Hooi, J.K.Y.; Lai, W.Y.; Ng, W.K.; Suen, M.M.Y.; Underwood, F.E.; Tanyingoh, D.; et al. Global Prevalence of Helicobacter pylori Infection: Systematic Review and Meta-Analysis. Gastroenterology 2017, 153, 420–429.
- Malaty, H.M.; Nyren, O. Epidemiology of Helicobacter pylori infection. Helicobacter 2003, 8 Suppl 1:8–12.
- Sjomina, O.; Pavlova, J.; Niv, Y.; Leja, M. Epidemiology of Helicobacter pylori infection. Helicobacter 2018, 23 Suppl 1:e12514.
- Roberts, S.E.; Morrison-Rees, S.; Samuel, D.G.; Thorne, K.; Akbari, A.; Williams, J.G. Review article: the prevalence of Helicobacter pylori and the incidence of gastric cancer across Europe. Aliment Pharmacol Ther 2016, 43, 334–345.
- Ruibys, G.; Denapiene, G.; Wright, R.A.; Irnius, A. Prevalence of Helicobacter Pylori Infection in Lithuanian Children. Off J Am Coll Gastroenterol | ACG 2004, 99, S32.
- Kupcinskas, J.; Leja, M. Management of Helicobacter pylori-Related Diseases in the Baltic States. Dig Dis 2014, 32, 295–301.
- Jonaitis, L.; Ivanauskas, A.; Janciauskas, D.; Funka, K.; Sudraba, A.; Tolmanis, I.; et al. Precancerous gastric conditions in high Helicobacter pylori prevalence areas: comparison between Eastern European (Lithuanian, Latvian) and Asian (Taiwanese) patients. Medicina (Kaunas) 2007, 43, 623-629.
- Boltin, D.; Niv, Y.; Schütte, K.; Schulz, C. Review: Helicobacter pylori and non‐malignant upper gastrointestinal diseases. Helicobacter 2019, 24 Suppl 1:e12637.
- Jonaitis, L.; Pellicano, R.; Kupcinskas, L. Helicobacter pylori and nonmalignant upper gastrointestinal diseases. Helicobacter 2018, 23 Suppl 1:e12522.
- Seid, A.; Tamir, Z.; Demsiss, W. Uninvestigated dyspepsia and associated factors of patients with gastrointestinal disorders in Dessie Referral Hospital, Northeast Ethiopia. BMC Gastroenterol 2018, 18, 13.
- Kabeer, K.K.; Ananthakrishnan, N.; Anand, C.; Balasundaram, S. Prevalence of Helicobacter Pylori Infection and Stress, Anxiety or Depression in Functional Dyspepsia and Outcome after Appropriate Intervention. J Clin Diagn Res 2017, 11, VC11–15.
- Tanaka, F.; Tominaga, K.; Fujikawa, Y.; Morisaki, T.; Otani, K.; Hosomi, S.; et al. Association between Functional Dyspepsia and Gastric Depressive Erosions in Japanese Intern Med 2019, 58, 321–328.
- Leung, W.K.; Chan, F.K.; Falk, M.S.; Suen, R.; Sung, J.J. Comparison of two rapid whole-blood tests for Helicobacter pylori infection in Chinese patients. J Clin Microbiol 1998, 36, 3441–3442.
- H. pylori Rapid Test Device (Whole Blood/Serum/Plasma) Package Insert. Available online: https://www.gimaitaly.com/DocumentiGIMA/Manuali/EN/M24528EN.pdf
- Leja, M.; Cine, E.; Rudzite, D.; Vilkoite, I.; Huttunen, T.; Daugule, I.; et al. Prevalence of Helicobacter pylori infection and atrophic gastritis in Latvia. Eur J Gastroenterol Hepatol 2012, 24, 1410–1417.
- Daugule, I.; Karklina, D.; Rudzite, D.; Remberga, S.; Rumba-Rozenfelde, I. Prevalence of Helicobacter pylori infection among preschool children in Latvia: no significant decrease in prevalence during a ten year period. Scand J Public Health 2016; 44; 418–422.
- Uibo, R.; Vorobjova, T.; Metsküla, K.; Kisand, K.; Wadström, T.; Kivik, T. Association of Helicobacter pylori and gastric autoimmunity: A population-based study. FEMS Immunology & Medical Microbiology 1995, 11, 65–68.
- Šebunova, N.; Štšepetova, J.; Sillakivi, T.; Mändar, R. The Prevalence of Helicobacter pylori in Estonian Bariatric Surgery Patients. Int J Mol Sci 2018, 19, 338.
- Tkachenko, M.A.; Zhannat, N.Z.; Erman, L.V.; Blashenkova, E.L.; Isachenko, S.V.; Isachenko, O.B.; et al. Dramatic changes in the prevalence of Helicobacter pylori infection during childhood: a 10-year follow-up study in Russia. J Pediatr Gastroenterol Nutr 2007, 45, 428–432.
- Mezmale, L.; Coelho, L.G.; Bordin, D.; Leja, M. Review: Epidemiology of Helicobacter pylori. Helicobacter 2020, 25 Suppl 1:e12734.
- Corojan, A.L.; Dumitrașcu, D.L.; Ciobanca, P.; Leucuta, D.C. Prevalence of Helicobacter pylori infection among dyspeptic patients in Northwestern Romania: A decreasing epidemiological trend in the last 30 years. Exp Ther Med 2020, 20, 3488–3492.
- Szaflarska-Popławska, A.; Soroczyńska-Wrzyszcz, A. Prevalence of Helicobacter pylori infection among junior high school students in Grudziadz, Poland. Helicobacter 2019, 24,
- Miyamoto, R.; Okuda, M.; Lin, Y.; Murotani, K.; Okumura, A.; Kikuchi, S. Rapidly decreasing prevalence of Helicobacter pylori among Japanese children and adolescents. J Infect Chemother 2019, 25, 526–530.
- Tang, M.Y.L.; Chung, P.H.Y.; Chan, H.Y.; Tam, P.K.H.; Wong, K.K. Recent trends in the prevalence of Helicobacter Pylori in symptomatic children: A 12-year retrospective study in a tertiary centre. J Pediatr Surg 2019, 54, 255–257.
- Almadi, M.A.; Aljebreen, A.M.; Tounesi, F.A.; Abdo, A.A. Helicobacter pylori prevalence among medical students in a high endemic area. Saudi Med J 2007, 28, 896–898.
- Al-Kadassy, A.; Suhail, M.; Baraheem, O. The Prevalence of Helicobacter Pylori Infection among Medical Sciences’ Students of Hodeidah University-Republic of Yemen. J High Inst Public Heal 2013, 43, 121–126.
- Muhsen, K.; Cohen, D.; Spungin-Bialik, A.; Shohat, T. Seroprevalence, correlates and trends of Helicobacter pylori infection in the Israeli population. Epidemiol Infect 2012, 140, 1207–1214.
- Seid, A.; Tamir, Z.; Demsiss, W. Uninvestigated dyspepsia and associated factors of patients with gastrointestinal disorders in Dessie Referral Hospital, Northeast Ethiopia. BMC Gastroenterol 2018, 18, 13.
- Tacikowski, T.; Bawa, S.; Gajewska, D.; Myszkowska-Ryciak, J.; Bujko, J.; Rydzewska, G. Current prevalence of Helicobacter pylori infection in patients with dyspepsia treated in Warsaw, Poland. Prz Gastroenterol 2017, 12, 135–139.
Additional information to the reviewer:
We would like to inform, that we have made some changes in the manuscript. The word "prevalence" has been changed to “seroprevalence”. This change may significantly influence the perception of the article.
The “prevalence” has been changed to “seroprevalence” in the title of the manuscript, the title of Figure 1, lines: 18, 31, 133, and 186, and additionally in all chapters and sentences where it is applicable.
For example, now the title of the manuscript is: “Changes in the Seroprevalence of Helicobacter pylori Among the Lithuanian Medical Students over the Last 25 Years and its Relation to Dyspeptic Symptoms“
Response to Reviewer 2 comments
Thank you very much for reviewing our manuscript. We appreciate your comments and suggestions. We hope that we have successfully addressed all of the concerns raised and we believe that the manuscript has been substantially improved. Our detailed responses to the comments and the description of the changes we have made to the manuscript are provided below.
Point 1: Methods - please indicate correctly the reference of the biological assays.
Response 1:
We added the reference for the “SureScreen Diagnostics Ltd” test:
“According to the manufacturer, the sensitivity of the “SureScreen Diagnostics Ltd” test is 87% and the specificity is 86% [21].”
- H. pylori Rapid Test Device (Whole Blood/Serum/Plasma) Package Insert https://www.gimaitaly.com/DocumentiGIMA/Manuali/EN/M24528EN.pdf
Point 2: Results - Figure 1: indicate in the same figure the evolution of the main symptoms.
Response 2:
Dear Reviewer,
We were thinking about the comprehensive representation of the symptoms before submitting the manuscript. However, we decided not to overload the manuscript with additional figures that are not statistically significant. This was not the primary aim of our study. In case of the figure with trendlines – it is too many trendlines in one figure and it would be difficult to add numbers and SD. Overall, it would be less informative and more difficult to read the figure. We would like to stress that the average intensity of symptoms was low and was not significantly different throughout the years. Moreover, we are emphasizing the difference of symptoms between HP-seropositive and HP-seronegative groups. We did not have an intension to compare the evolution of the symptoms in different years. Therefore, we decided to represent the prevalence and intensity of symptoms in the Tables 2 and 3, as it looked most informative.
We hope you could accept our explanation and decision.
Table 2. Prevalence of upper gastrointestinal tract symptoms in H. pylori-seropositive and H. pylori-seronegative students
|
|
Study year |
|
|||||||
|
|
2012 |
2016 |
2020 |
||||||
|
Dyspeptic symptoms |
HP+ |
HP- |
p |
HP+ |
HP- |
p |
HP+ |
HP- |
p |
|
Epigastric discomfort |
47.4% |
41.5% |
>0.05 |
47.8% |
40.5% |
>0.05 |
28.6% |
40.8% |
>0.05 |
|
Heartburn |
29.8% |
22.3% |
>0.05 |
40.6% |
33.7% |
>0.05 |
23.8% |
28.0% |
>0.05 |
|
Regurgitation |
22.8% |
16.2% |
>0.05 |
20.3% |
27.4% |
>0.05 |
23.8% |
17.6% |
>0.05 |
|
Hunger-like pain |
66.7% |
58.5% |
>0.05 |
58.0% |
66.8% |
>0.05 |
66.7% |
65.6% |
>0.05 |
|
Nausea |
28.1% |
30.8% |
>0.05 |
46.4% |
40.5% |
>0.05 |
38.1% |
43.2% |
>0.05 |
|
Borborygmus |
75.4% |
76.7% |
>0.05 |
85.5% |
79.5% |
>0.05 |
66.7% |
73.6% |
>0.05 |
|
Epigastric fullness |
54.4% |
59.2% |
>0.05 |
63.8% |
63.2% |
>0.05 |
38.1% |
58.9% |
>0.05 |
|
Belching |
38.6% |
44.6% |
>0.05 |
58.0% |
57.4% |
>0.05 |
28.6% |
53.2% |
0.037 |
Table 3. Intensity of upper gastrointestinal tract symptoms in H. pylori-seropositive and H. pylori-seronegative
students
|
|
Study year |
|
|||||||
|
|
2012 |
2016 |
2020 |
||||||
|
Dyspeptic symptoms |
HP+ |
HP- |
p |
HP+ |
HP- |
p |
HP+ |
HP- |
p |
|
Epigastric discomfort |
1.1±1.4 |
0.7±1.0 |
>0.05 |
2.0±1.3 |
2.0±1.5 |
>0.05 |
2.5±1.5 |
2.1±1.4 |
>0.05 |
|
Heartburn |
0.7±1.3 |
0.4±0.9 |
>0.05 |
2.3±4.1 |
1.8±1.3 |
>0.05 |
2.0±1.4 |
2.3±1.6 |
>0.05 |
|
Regurgitation |
0.4±0.8 |
0.3±0.8 |
>0.05 |
1.4±1.0 |
1.5±1.1 |
>0.05 |
1.4±0.9 |
1.7±1.2 |
>0.05 |
|
Hunger-like pain |
1.5±1.4 |
1.3±1.5 |
>0.05 |
2.2±1.4 |
2.5±1.4 |
>0.05 |
1.6±0.9 |
2.1±1.1 |
>0.05 |
|
Nausea |
0.5±1.2 |
0.7±1.2 |
>0.05 |
2.0±1.2 |
2.0±1.5 |
>0.05 |
1.6±0.7 |
2.2±1.5 |
>0.05 |
|
Borborygmus |
1.4±1.2 |
1.8±1.5 |
>0.05 |
3.2±1.6 |
2.9±1.4 |
>0.05 |
1.9±1.2 |
2.5±1.3 |
>0.05 |
|
Epigastric fullness |
1.1±1.3 |
1.3±1.4 |
>0.05 |
2.6±1.7 |
2.6±1.6 |
>0.05 |
2.3±1.3 |
2.2±1.3 |
>0.05 |
|
Belching |
0.6±1.0 |
0.8±1.2 |
>0.05 |
2.2±1.5 |
2.1±1.3 |
>0.05 |
1.5±0.5 |
1.7±1.1 |
>0.05 |
Point 3: Discussion - Numerous sentence lack reference. Please add.
Response 3:
Dear Reviewer, of course, we may agree that there is not enough references in the “Discussion” chapter. We tried to support our main discussion points with the appropriate references. However, there is a recommendation to limit the list of references to the reasonable amount. Therefore, we could not be very extensive.
According to Your suggestion, we have added some references and added the numbers of the references to some sentences.
Probably, we may kindly add few more references, in case the reviewer could indicate the points where the references should be added? We are not sure in which parts of the “Discussion” the references are insufficient as the Reviewer did not indicate specifically.
Please find below the updated chapter “Discussion” and the reference list, with the highlighted changes
- Discussion
Our results indicate the obvious decline in the seroprevalence of H. pylori among the students of LUHS during the last 25 years. There is a clear lack of large-scale epidemiological data on the prevalence of H. pylori in neighboring countries as well as in whole Eastern-Central European region. Epidemiological studies in recent years have shown the different prevalence of H. pylori in the neighboring countries [22-25,27,28]. It has been stated that the prevalence of H. pylori among Latvian adults (n=3564) was 79.2% in the year 2011 [22] and in fact there was no significant decrease of this infection among Latvian children during a 10-year period (2000-2010) [23]. A population study (n=1461) was performed in Estonia in the year 1993 and revealed seroprevalence of H. pylori infection in 87% of the participants [24]. More recent studies have shown the prevalence of H. pylori 69% in Tartu’s population [14] and 64.7% among the Estonian bariatric surgery patients in the year 2018 [25]. In Estonian children, the H. pylori seroprevalence rate was 42% in 1991 and 28.1% in 2002 [25]. Review article [11] and other studies concluded that the prevalence of H. pylori ranges from 13% in children [26] to 65.6% in adults [27] in Russia, is around 32% in Hungary [27], 23.5% in Czech Republic, 35.8% in Poland [10] and 40.8% in Romania [28].
There are very few studies available, testing participants of the similar cohorts (similar age groups (~20-25 years old) and of the same contingent) as our current research [27,29,31-34]. It has been stated that the prevalence of H. pylori was 23.6% among junior high school students in Poland [29], 44.1% in the age group of 18-24 years in Russia [27], ranges from 7% up to 15% in Japan [30], was 14.3% in China [31], 35% among Saudi Arabia medical students [32], 68% among Yemen medical students [33], 54.7% among Taiwan high-school students and 42% in young Jewish population [34]. Most of the above-mentioned studies have analyzed and compared data from various years and have clearly stated that the prevalence of H. pylori infection is decreasing. When compared to the studies on the same age groups in different countries, we can state that the prevalence of H. pylori in Lithuania is similar to Poland, Japan or China [29-31].
One of the main advantages of our research is the fact that all of the four groups of patients from different time periods were tested using the same methodology (finger capillary blood serology) and evaluated the similar cohort (populations of similar mean age in the same university). Such studies are hardly available in other countries.
However, there are some drawbacks in our research, which need to be considered. We have to recognize, that the cohorts of the four analyzed studies are not large enough in order to evaluate the general epidemiological situation in Lithuania but at least we can make an assumption that it should correspond to the situation in other countries. In addition, the non-invasive diagnosis of H. pylori serologic tests from finger capillary blood has been used. The current edition of Maastricht V/Florence guidelines on the management of H. pylori infection, which is being used in European countries, recommended urea breath test (UBT) or monoclonal stool antigen (SAT) test for the non-invasive diagnosis of H. pylori and rapid serology tests are not the best choice. However, the same guidelines state that validated serological tests with the sensitivity and specificity above 90% can be used for the non-invasive diagnostics [2].
In general, we did not find significant differences in the prevalence and intensity of dyspeptic symptoms among H. pylori-positive and negative students, except for the symptom of belching which was more prevalent in H. pylori-negative students in the year 2020. We would like to note that the average intensities of symptoms were quite low (1-2 points), which means that the symptoms were not really bothering and were very accidental. We assume that it is the reason, why there were no differences between H. pylori-positive and negative groups. In studies, that reported the relation between H. pylori infection and dyspeptic symptoms, usually the patients with bothering dyspeptic symptoms were investigated [35,36].
- Conclusions
To conclude our study, the seroprevalence for Helicobacter pylori was established in 14.2% of students of Lithuanian University of Health Sciences in the year 2020. Over the last 25 years, the seroprevalence of H. pylori among the students of LUHS has decreased significantly. No consistent differences in dyspeptic symptoms among H. pylori seropositive and -negative subgroups were found.
References
- Wroblewski, L.E.; Peek, R.M.J.; Wilson, K.T. Helicobacter pylori and gastric cancer: factors that modulate disease risk. Clin Microbiol Rev 2010, 23, 713–739.
- Malfertheiner, P.; Megraud, F.; O’Morain, C.A.; Gisbert, J.P.; Kuipers, E.J.; Axon, A.T.; et al. Management of Helicobacter pylori infection - the Maastricht V/Florence Consensus Report. Gut 2017, 66, 6–30.
- Chey, W.D.; Leontiadis, G.I.; Howden, C.W.; Moss, S.F. Correction: ACG Clinical Guideline: Treatment of Helicobacter pylori Infection. The American journal of gastroenterology 2018, 113, 1102.
- Kato, M.; Ota, H.; Okuda, M.; Kikuchi, S.; Satoh, K.; Shimoyama, T.; et al. Guidelines for the management of Helicobacter pylori infection in Japan: 2016 Revised Edition. Helicobacter 2019, 24, e12597.
- Moss, S.F. The Clinical Evidence Linking Helicobacter pylori to Gastric Cancer. Cell Mol Gastroenterol Hepatol 2017, 3, 183–191.
- Šterbenc, A.; Jarc, E.; Poljak, M.; Homan, M. Helicobacter pylori virulence genes. World J Gastroenterol 2019, 25, 4870–4884.
- Kupcinskas, L.; Rasmussen, L.; Jonaitis, L.; Kiudelis, G.; Jørgensen, M.; Urbonaviciene, N.; et al. Evolution of Helicobacter pylori susceptibility to antibiotics during a 10-year period in Lithuania. APMIS 2013, 121, 431–436.
- Hooi, J.K.Y.; Lai, W.Y.; Ng, W.K.; Suen, M.M.Y.; Underwood, F.E.; Tanyingoh, D.; et al. Global Prevalence of Helicobacter pylori Infection: Systematic Review and Meta-Analysis. Gastroenterology 2017, 153, 420–429.
- Malaty, H.M.; Nyren, O. Epidemiology of Helicobacter pylori infection. Helicobacter 2003, 8 Suppl 1:8–12.
- Sjomina, O.; Pavlova, J.; Niv, Y.; Leja, M. Epidemiology of Helicobacter pylori infection. Helicobacter 2018, 23 Suppl 1:e12514.
- Roberts, S.E.; Morrison-Rees, S.; Samuel, D.G.; Thorne, K.; Akbari, A.; Williams, J.G. Review article: the prevalence of Helicobacter pylori and the incidence of gastric cancer across Europe. Aliment Pharmacol Ther 2016, 43, 334–345.
- Ruibys, G.; Denapiene, G.; Wright, R.A.; Irnius, A. Prevalence of Helicobacter Pylori Infection in Lithuanian Children. Off J Am Coll Gastroenterol | ACG 2004, 99, S32.
- Kupcinskas, J.; Leja, M. Management of Helicobacter pylori-Related Diseases in the Baltic States. Dig Dis 2014, 32, 295–301.
- Jonaitis, L.; Ivanauskas, A.; Janciauskas, D.; Funka, K.; Sudraba, A.; Tolmanis, I.; et al. Precancerous gastric conditions in high Helicobacter pylori prevalence areas: comparison between Eastern European (Lithuanian, Latvian) and Asian (Taiwanese) patients. Medicina (Kaunas) 2007, 43, 623-629.
- Boltin, D.; Niv, Y.; Schütte, K.; Schulz, C. Review: Helicobacter pylori and non‐malignant upper gastrointestinal diseases. Helicobacter 2019, 24 Suppl 1:e12637.
- Jonaitis, L.; Pellicano, R.; Kupcinskas, L. Helicobacter pylori and nonmalignant upper gastrointestinal diseases. Helicobacter 2018, 23 Suppl 1:e12522.
- Seid, A.; Tamir, Z.; Demsiss, W. Uninvestigated dyspepsia and associated factors of patients with gastrointestinal disorders in Dessie Referral Hospital, Northeast Ethiopia. BMC Gastroenterol 2018, 18, 13.
- Kabeer, K.K.; Ananthakrishnan, N.; Anand, C.; Balasundaram, S. Prevalence of Helicobacter Pylori Infection and Stress, Anxiety or Depression in Functional Dyspepsia and Outcome after Appropriate Intervention. J Clin Diagn Res 2017, 11, VC11–15.
- Tanaka, F.; Tominaga, K.; Fujikawa, Y.; Morisaki, T.; Otani, K.; Hosomi, S.; et al. Association between Functional Dyspepsia and Gastric Depressive Erosions in Japanese Intern Med 2019, 58, 321–328.
- Leung, W.K.; Chan, F.K.; Falk, M.S.; Suen, R.; Sung, J.J. Comparison of two rapid whole-blood tests for Helicobacter pylori infection in Chinese patients. J Clin Microbiol 1998, 36, 3441–3442.
- pylori Rapid Test Device (Whole Blood/Serum/Plasma) Package Insert. Available online: https://www.gimaitaly.com/DocumentiGIMA/Manuali/EN/M24528EN.pdf
- Leja, M.; Cine, E.; Rudzite, D.; Vilkoite, I.; Huttunen, T.; Daugule, I.; et al. Prevalence of Helicobacter pylori infection and atrophic gastritis in Latvia. Eur J Gastroenterol Hepatol 2012, 24, 1410–1417.
- Daugule, I.; Karklina, D.; Rudzite, D.; Remberga, S.; Rumba-Rozenfelde, I. Prevalence of Helicobacter pylori infection among preschool children in Latvia: no significant decrease in prevalence during a ten year period. Scand J Public Health 2016; 44; 418–422.
- Uibo, R.; Vorobjova, T.; Metsküla, K.; Kisand, K.; Wadström, T.; Kivik, T. Association of Helicobacter pylori and gastric autoimmunity: A population-based study. FEMS Immunology & Medical Microbiology 1995, 11, 65–68.
- Šebunova, N.; Štšepetova, J.; Sillakivi, T.; Mändar, R. The Prevalence of Helicobacter pylori in Estonian Bariatric Surgery Patients. Int J Mol Sci 2018, 19, 338.
- Tkachenko, M.A.; Zhannat, N.Z.; Erman, L.V.; Blashenkova, E.L.; Isachenko, S.V.; Isachenko, O.B.; et al. Dramatic changes in the prevalence of Helicobacter pylori infection during childhood: a 10-year follow-up study in Russia. J Pediatr Gastroenterol Nutr 2007, 45, 428–432.
- Mezmale, L.; Coelho, L.G.; Bordin, D.; Leja, M. Review: Epidemiology of Helicobacter pylori. Helicobacter 2020, 25 Suppl 1:e12734.
- Corojan, A.L.; Dumitrașcu, D.L.; Ciobanca, P.; Leucuta, D.C. Prevalence of Helicobacter pylori infection among dyspeptic patients in Northwestern Romania: A decreasing epidemiological trend in the last 30 years. Exp Ther Med 2020, 20, 3488–3492.
- Szaflarska-Popławska, A.; Soroczyńska-Wrzyszcz, A. Prevalence of Helicobacter pylori infection among junior high school students in Grudziadz, Poland. Helicobacter 2019, 24,
- Miyamoto, R.; Okuda, M.; Lin, Y.; Murotani, K.; Okumura, A.; Kikuchi, S. Rapidly decreasing prevalence of Helicobacter pylori among Japanese children and adolescents. J Infect Chemother 2019, 25, 526–530.
- Tang, M.Y.L.; Chung, P.H.Y.; Chan, H.Y.; Tam, P.K.H.; Wong, K.K. Recent trends in the prevalence of Helicobacter Pylori in symptomatic children: A 12-year retrospective study in a tertiary centre. J Pediatr Surg 2019, 54, 255–257.
- Almadi, M.A.; Aljebreen, A.M.; Tounesi, F.A.; Abdo, A.A. Helicobacter pylori prevalence among medical students in a high endemic area. Saudi Med J 2007, 28, 896–898.
- Al-Kadassy, A.; Suhail, M.; Baraheem, O. The Prevalence of Helicobacter Pylori Infection among Medical Sciences’ Students of Hodeidah University-Republic of Yemen. J High Inst Public Heal 2013, 43, 121–126.
- Muhsen, K.; Cohen, D.; Spungin-Bialik, A.; Shohat, T. Seroprevalence, correlates and trends of Helicobacter pylori infection in the Israeli population. Epidemiol Infect 2012, 140, 1207–1214.
- Seid, A.; Tamir, Z.; Demsiss, W. Uninvestigated dyspepsia and associated factors of patients with gastrointestinal disorders in Dessie Referral Hospital, Northeast Ethiopia. BMC Gastroenterol 2018, 18, 13.
- Tacikowski, T.; Bawa, S.; Gajewska, D.; Myszkowska-Ryciak, J.; Bujko, J.; Rydzewska, G. Current prevalence of Helicobacter pylori infection in patients with dyspepsia treated in Warsaw, Poland. Prz Gastroenterol 2017, 12, 135–139.
Additional information to the reviewer:
We would like to inform, that we have made some changes in the manuscript. The word "prevalence" has been changed to “seroprevalence”. This change may significantly influence the perception of the article.
The “prevalence” has been changed to “seroprevalence” in the title of the manuscript, the title of Figure 1, lines: 18, 31, 133, and 186, and additionally in all chapters and sentences where it is applicable.
For example, now the title of the manuscript is: “Changes in the Seroprevalence of Helicobacter pylori Among the Lithuanian Medical Students over the Last 25 Years and its Relation to Dyspeptic Symptoms“
Round 2
Reviewer 1 Report
Thank you to the Authors for following most of my suggestions. I believe that the quality of the maniscript has increased